# The genetic requirements of fatty acid import by *Mycobacterium tuberculosis* within macrophages

Evgeniya V Nazarova, Christine R Montague, Lu Huang, Thuy La, David Russell, Brian C VanderVen*

Department of Microbiology and Immunology, College of Veterinary Medicine, Cornell University, Ithaca, United States

**Abstract** *Mycobacterium tuberculosis* (Mtb) imports and metabolizes fatty acids to maintain infection within human macrophages. Although this is a well-established paradigm, the bacterial factors required for fatty acid import are poorly understood. Previously, we found that LucA and Mce1 are required for fatty acid import in Mtb (Nazarova et al., 2017). Here, we identified additional Mtb mutants that have a reduced ability to import a fluorescent fatty acid substrate during infection within macrophages. This screen identified the novel genes as *rv2799* and *rv0966c* as be necessary for fatty acid import and confirmed the central role for Rv3723/LucA and putative components of the Mce1 fatty acid transporter (Rv0200/OmamB, Rv0172/Mce1D, and Rv0655/MceG) in this process.
DOI: https://doi.org/10.7554/eLife.43621.001

## Introduction

Multiple lines of evidence indicate that fatty acids are utilized by *Mycobacterium tuberculosis* (Mtb) during infection. Early studies established that *Mycobacterium tuberculosis* (Mtb) isolated from mouse lungs preferentially metabolizes fatty acids ex vivo (*Bloch and SEGAL, 1956*). It is known that Mtb upregulates genes needed for fatty acid utilization during infection in macrophages, mouse challenge models, and within human lung tissues (*Fontán et al., 2008*; *Homolka et al., 2010*; *Rachman et al., 2006*; *Rohde et al., 2007*; *Rohde et al., 2012*; *Schnappinger et al., 2003*; *Tailleux et al., 2008*). Functional experiments have also established that Mtb can import and metabolize fatty acids (*Lee et al., 2013*) or the acyl-chains derived from triacylglycerol (*Daniel et al., 2011*), and fluorescent fatty acid trafficking studies have visualized the presence of lipid inclusions in intracellular Mtb (*Podinovskaia et al., 2013*).

TB disease pathology may also contribute to Mtb's metabolic preference for lipids given that the bacterium resides within tissue environments that are laden with lipid substrates. In human granulomatous lesions, intracellular Mtb resides in foamy macrophages (*Peyron et al., 2008*) and these cells are known to concentrate triacylglycerol, cholesterol, and cholesterol esters (*Almeida et al., 2012*; *McClean and Tobin, 2016*). Additionally, the caseous centers of necrotic TB lesions are enriched in these same lipids and extracellular bacteria are commonly found embedded in this nutrient-rich material (*Cáceres et al., 2009*; *Hunter et al., 2007*; *Kim et al., 2010*).

Although the principles of fatty acid uptake in Mtb are poorly understood, we recently discovered that Rv3723/LucA and the Mce1 transporter complex are required for fatty acid import (*Nazarova et al., 2017*). Given that these proteins are needed for full Mtb virulence in murine infection models (*Joshi et al., 2006*; *Nazarova et al., 2017*), we sought to identify additional genes that are involved in this process. Here, we use a forward genetic screen and a fluorescence-activated cell sorting (FACS) approach to identify Mtb mutants deficient in Bodipy-palmitate import during

*For correspondence:
bcv8@cornell.edu

Competing interests: The authors declare that no competing interests exist.

macrophage infection. This approach identified novel genes (*rv2799*, *rv0966c*, *rv0655*/*mceG*, and *rv0200*/*omamB*) and confirmed that genes encoding putative components the Mce1 transporter and Rv3723/LucA facilitate fatty acid import in Mtb. Together these data support our previous findings and are consistent with the concept that fatty acid import in Mtb is mediated by a network of shared and dedicated proteins that import fatty acids via Mce transporter in Mtb.

## Results

### Genetic approach to identify fatty acid import mutants

Rv3723/LucA is required for both fatty acid and cholesterol import in Mtb (*Nazarova et al., 2017*), and we have demonstrated that a robust phenotype of a Mtb mutant lacking Rv3723/LucA (Δ*lucA*:: hyg) is the inability of this strain to import the fluorescent fatty acid substrate, Bodipy-palmitate during infection in macrophages. Since this trait is easily quantifiable by flow cytometric analysis we developed a FACS-based assay to isolate additional mutants that are defective in Bodipy-palmitate import. For this FACS-based screen, we infected macrophages with a Mtb strain that constitutively expresses mCherry and at 3 days post-infection the macrophages were pulse labeled with Bodipy-palmitate. Following the Bodipy-palmitate labeling procedure, the infected macrophages were lysed and intracellular bacteria were isolated on a sucrose cushion for flow cytometry analysis or sorting based on Bodipy-palmitate incorporation into the bacterial cells (*Figure 1A*) (*Nazarova et al., 2018*).

We optimized this screening approach using wild type (WT) and Δ*lucA*::hyg Mtb strains that express mCherry to establish our workflow. Briefly, we infected macrophages with each strain, pulse-labeled infected cells with the fluorescent substrate, and measured bacterial incorporation of Bodipy-palmitate by flow cytometry. These strains allowed us to establish a sorting method that isolated mCherry-positive singlet bacteria that are Bodipy-low. Analysis of lysates from uninfected macrophages confirmed that this methodology exclusively captured bacterial cells (*Figure 1B and C*). Applying this sorting strategy to the Δ*lucA*::hyg mutant confirmed that this strain incorporated less Bodipy-palmitate relative to WT (*Figure 1D*). For further validation of this FACS-based screening approach, we isolated WT and Δ*lucA*::hyg bacteria from Bodipy-palmitate pulse-labeled macrophages and combined these bacterial populations at a ratio of 1:1. We FACS sorted this mixed bacterial sample into Bodipy-low (*Figure 1B*) and excluded the Bodipy-high populations (*Figure 1D*). Flow analysis determined that sorted bacterial Bodipy-low population approached ~84% purity (*Figure 1E*). Using the hygromycin-resistance marker carried by the Δ*lucA*::hyg mutant, we evaluated the specificity of this sorting procedure by plating on selective agar. Of the sorted cells in the desired Bodipy-low population 98.68 ± 1.31% of the recovered colonies were hygromycin-resistant, demonstrating that the majority of sorted population were indeed Δ*lucA*::hyg bacteria indicating the high degree of specificity with this approach.

### Isolating and identifying fatty acid import mutants

Using this FACS-based approach, we sought to isolate mutants with a reduced ability to import Bodipy-palmitate. Using the MycoMarT7 transposon (*Sassetti et al., 2001*) delivered by phage transduction (*Bardarov et al., 1997*), we generated a transposon mutant library (~$2\times10^4$ mutants) in a Mtb Erdman strain which expresses mCherry from a chromosomally integrated plasmid (*Stover et al., 1991*). For screening, macrophages were infected with the mutant library, pulse labeled with the fluorescent substrate, and the Bodipy-low bacteria were isolated by FACS. On the day of the experiment, sorting gates were established using WT and Δ*lucA*::hyg bacteria isolated from Bodipy-palmitate-labeled macrophages. Events falling within Bodipy-low gate were sorted and plated onto selective media. Of the $10^5$ events collected in the Bodipy-low sort, only ~12.8% (~$1.28\times10^4$) gave rise to colonies. For comparison, when the Bodipy-low population was sorted from the 1:1 mixture containing WT and Δ*lucA*::hyg strains ~ 71.9 ± 0.67% of the sorted events produced colonies. The reduced recovery of mutant bacteria from the Bodipy-low population suggests that a large proportion (~60%) of the clones are non-viable or non-culturable with our plating conditions. To isolate mutants with a strong Bodipy-palmitate import defect, we employed an enrichment strategy.

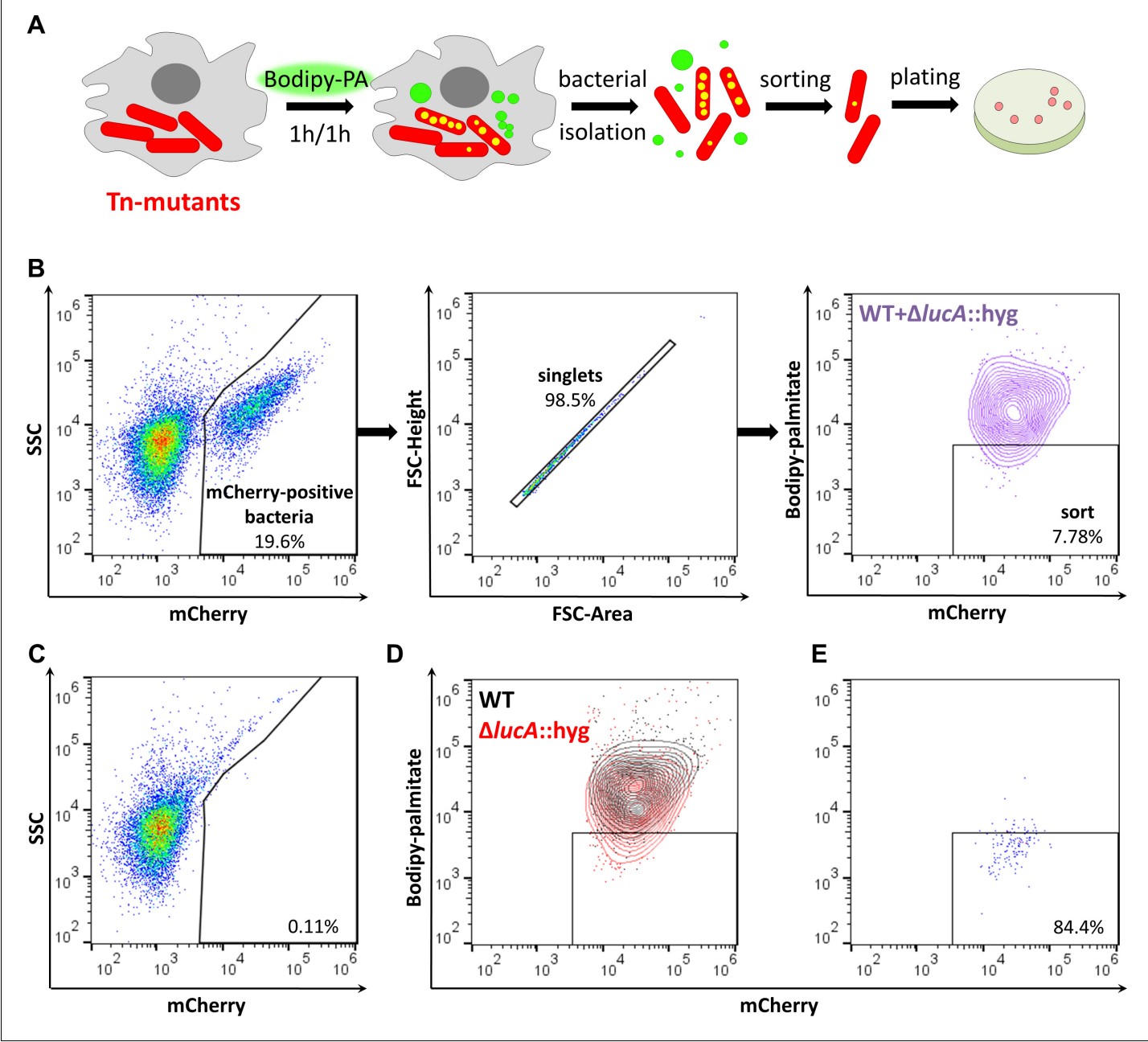

**Figure 1.** Developing a screening approach to identify fatty acid import mutants in Mtb. (**A**) Schematic of the genetic screen. Macrophages are infected with mCherry-Mtb for three days. Intracellular transposon mutant library (Tn-mutants) is pulse labeled with Bodipy-palmitate (Bodipy-PA). Intracellular bacteria are isolated, sorted, and plated on selective agar. Transposon insertion sites were determined for the mutant pools recovered from agar. (**B**) Gating strategy to isolate single Bodipy-low Mtb cells. (**C**) Analysis of uninfected macrophage lysates that have been pulse labeled with Bodipy-palmitate. (**D**) Flow analysis of a 1:1 mixture containing mCherry-positive WT and Δ*lucA*::hyg bacteria. (**E**) Purity of sorted low Bodipy-low population.
DOI: https://doi.org/10.7554/eLife.43621.002

Following the second enrichment sort, $10^5$ Bodipy-low events were collected, and ~$5.3\times10^4$ (53%) of this population yielded viable colonies. Each round of selection enriched for mutants with a decreased ability to incorporate Bodipy-palmitate (*Figure 2A*).

We analyzed the pools of sorted mutants harvested after first and second rounds of selection and compared this data to the input pool (original library) using a Transposon Site Hybridization (TraSH) method (*Figure 2—source data 1*) (*Lee et al., 2013*; *Murry et al., 2008*) (*Lee et al., 2013*;

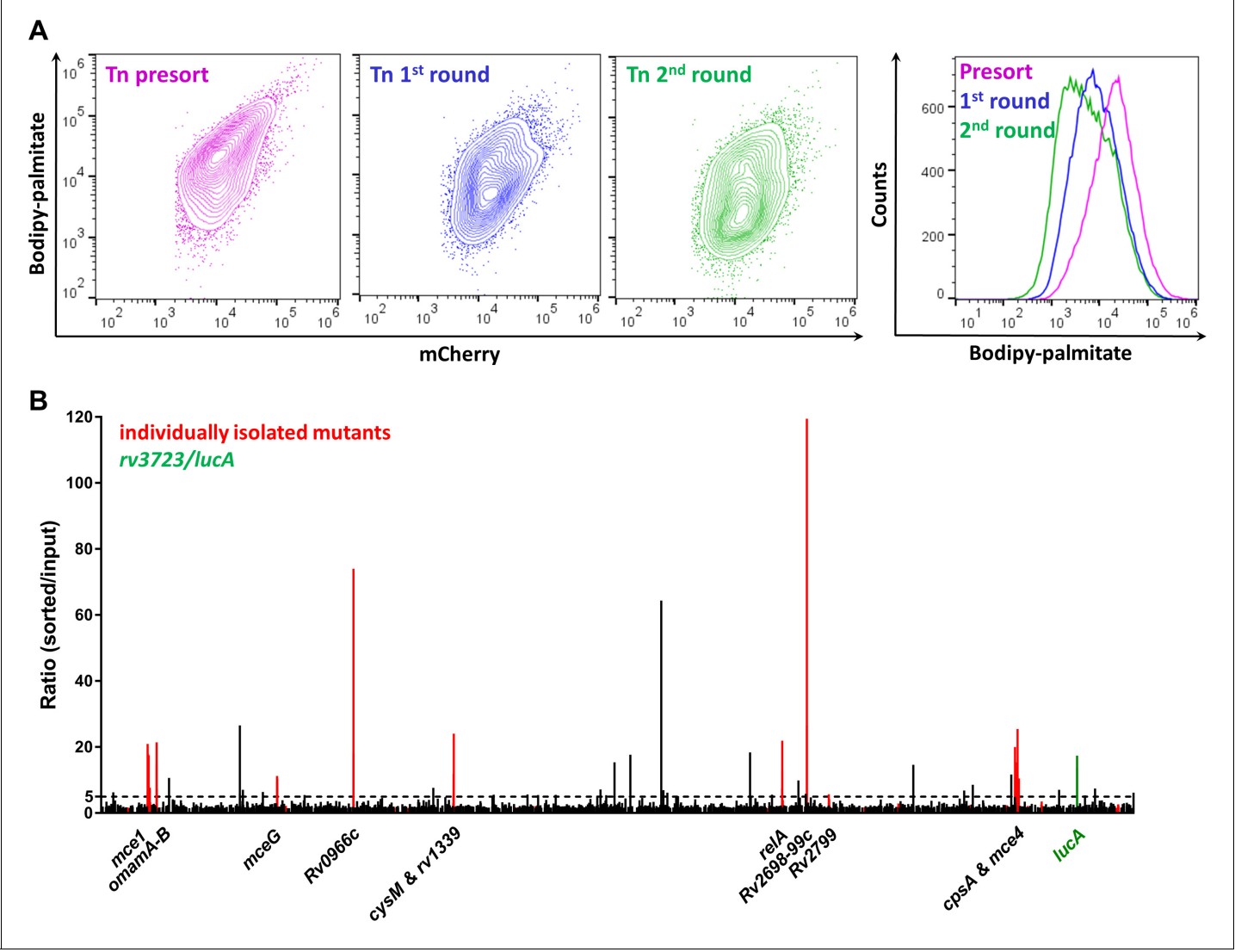

**Figure 2.** Enrichment and identification of mutants with a reduced ability to import Bodipy-palmitate. (**A**) Flow analysis of Bodipy-palmitate incorporation by input mutant pool (Tn presort) and subsequent enriched mutant pools. (**B**) Relative abundance of transposon insertion sites following the second enrichment sort which are overrepresented >1.5 fold with p < 0.05. The x-axis represents position of the transposon insertion in the Mtb chromosome and peak height represents relative level of overrepresentation in the analysis. Individually isolated mutants are indicated in red. The *rv3723/lucA* mutation is highlighted in green and the dotted line indicates fivefold cut-off. Full gene lists and probe sequences used in the TraSH analysis are supplied in *Figure 2—source data 1*. Fold-change distribution of insertion sites shown on *Figure 2B* (>1.5 fold overrepresented with p < 0.05 after two rounds of enrichment) among the first and second rounds of sorting is supplied in *Figure 2—figure supplement 1*. A summary of the isolated mutants is provided in *Figure 2—source data 2*.
DOI: https://doi.org/10.7554/eLife.43621.003

The following source data and figure supplement are available for figure 2:

**Source data 1.** Summary of TraSH analysis results.
DOI: https://doi.org/10.7554/eLife.43621.005

**Source data 2.** Spreadsheet of all individually isolated mutants.
DOI: https://doi.org/10.7554/eLife.43621.006

**Figure supplement 1.** Fold-change distribution of insertion sites from the enrichment screens.
DOI: https://doi.org/10.7554/eLife.43621.004

*Murry et al., 2008*). This analysis quantifies the relative abundance of transposon site insertions in the chromosomes of the mutants isolated following each round of enrichment. The curated list of transposon insertion sites overrepresented from each enrichment screen is provided (*Figure 2—source data 1*). Our analysis of the mutants isolated from the second enrichment sort identified 73 insertion sites that map to 65 genes or intragenic regions with a fivefold overrepresentation with p-value < 0.05 (*Figure 2—figure supplement 1*; *Figure 2—source data 1*). Importantly, our selection strategy also confirmed that transposon mutations in *rv3723/lucA* are overrepresented following the second round of enrichment (13.67-fold overrepresented, p = $1.24 \times 10^{-11}$) (*Figure 2B* and *Figure 2—source data 2*).

To determine if our TraSH-based predictions correlate with an inability to import fatty acids, we collected individual clones from each enrichment pool and mapped the transposon insertion before phenotypically confirming the clones. For this analysis, 38 and 46 independent clones were isolated from the first and the second rounds of selection, respectively. DNA sequencing confirmed that we selected both sibling mutants and clones with independent insertions in the same genes from the FACS-based enrichment process (*Figure 2—source data 2*). Thirty-seven of the 46 clones (80%) isolated in the second round of selection contained insertions in the same genes as mapped by TraSH (>5 fold overrepresentation; p < 0.05) revealing a significant correlation in the dataset (*Figure 2B* and *Figure 2—source data 2*).

## Confirmation of mutant phenotypes

We next quantified Bodipy-palmitate import in individual Mtb mutants during infection in macrophages using flow cytometry. In total, 13 transposon mutants with mutations overrepresented >5 fold in our TraSH analysis were selected (*Figure 2—source data 2*). Bodipy-palmitate import was quantified in these mutants during infection in macrophages and import of the fluorescent substrate was normalized by assigning 100% import to WT and 0% import to the ΔlucA::hyg mutant (*Figure 3A,X* axis). This analysis determined that 6 of the 13 mutants were defective in Bodipy-palmitate import at levels less than 50%. This flow cytometric analysis was confirmed by confocal microscopy demonstrating that the same six mutants have the strongest defect in Bodipy-palmitate uptake within macrophages (*Figure 3—figure supplement 1*).

To determine if fatty acid assimilation defect in these mutants is induced by macrophage environment, we next quantified the metabolism of radiolabeled fatty acids when the bacteria were cultured in axenic media. Because fatty acid catabolism is an intracellular process, we used the catabolic release of $^{14}C$-$CO_2$ from [$1$-$^{14}C$]-oleate as an indirect measure of fatty acid import in Mtb (*Figure 3A,Y* axis). Mutants defective in fatty acid import during infection in macrophages that demonstrated a pronounced reduction in $^{14}C$-fatty acid metabolism had transposon insertions in genes encoded within the *mce1* operon (*rv0172/mce1D*) or in genes encoding proteins believed to be part of the Mce1 transporter (*rv0655/mceG, rv0200/omamB*) (*Joshi et al., 2006*; *Perkowski et al., 2016*) indicating an important role for these genes and Mce1 complex in the import and/or metabolism of fatty acids. A mutant with transposon insertion in *rv0966c*, a gene of unknown function also has both a reduced ability to import Bodipy-palmitate in macrophages (~85% reduction) and a decreased ability to metabolize $^{14}C$-fatty acid in liquid culture (~60% reduction) (*Figure 3A*).

Interestingly, several transposon mutants have phenotypes that are condition or substrate dependent. One mutant with a transposon insertion within the *mce1* operon (*rv0175/mam1A*) has a minor defect in Bodipy-palmitate import during infection in macrophages (~20% reduction) but has pronounced defect in metabolizing $^{14}C$-fatty acid in liquid culture (~98% reduction) (*Figure 3A*). Mutants with transposon insertions in *rv2799* and *rv2583c/relA* have a decreased ability to import Bodipy-palmitate in macrophages (~55% and~50% reduction, respectively) while both of these mutants displayed an increase in the bacterium's ability to metabolize $^{14}C$-fatty acid in liquid culture (~35% and~125% increase, respectively) (*Figure 3A*). A *rv3484/cpsA* transposon mutant has a decreased ability to import Bodipy-palmitate in macrophages (~70% reduction), yet this mutant displayed a subtle decrease in the bacterium's ability to metabolize $^{14}C$-fatty acid in liquid culture (~20% decrease)

Because several metabolic processes can decrease the catabolic release of $^{14}C$-$CO_2$ from [$1$-$^{14}C$]-oleate we directly quantified [$1$-$^{14}C$]-oleate import when the bacteria are cultured in axenic media (*Nazarova et al., 2017*). Quantification of fatty acid import in four transposon mutants deficient in metabolism of [$^{14}C$]-oleate confirmed that these clones also had a statistically significant decrease in

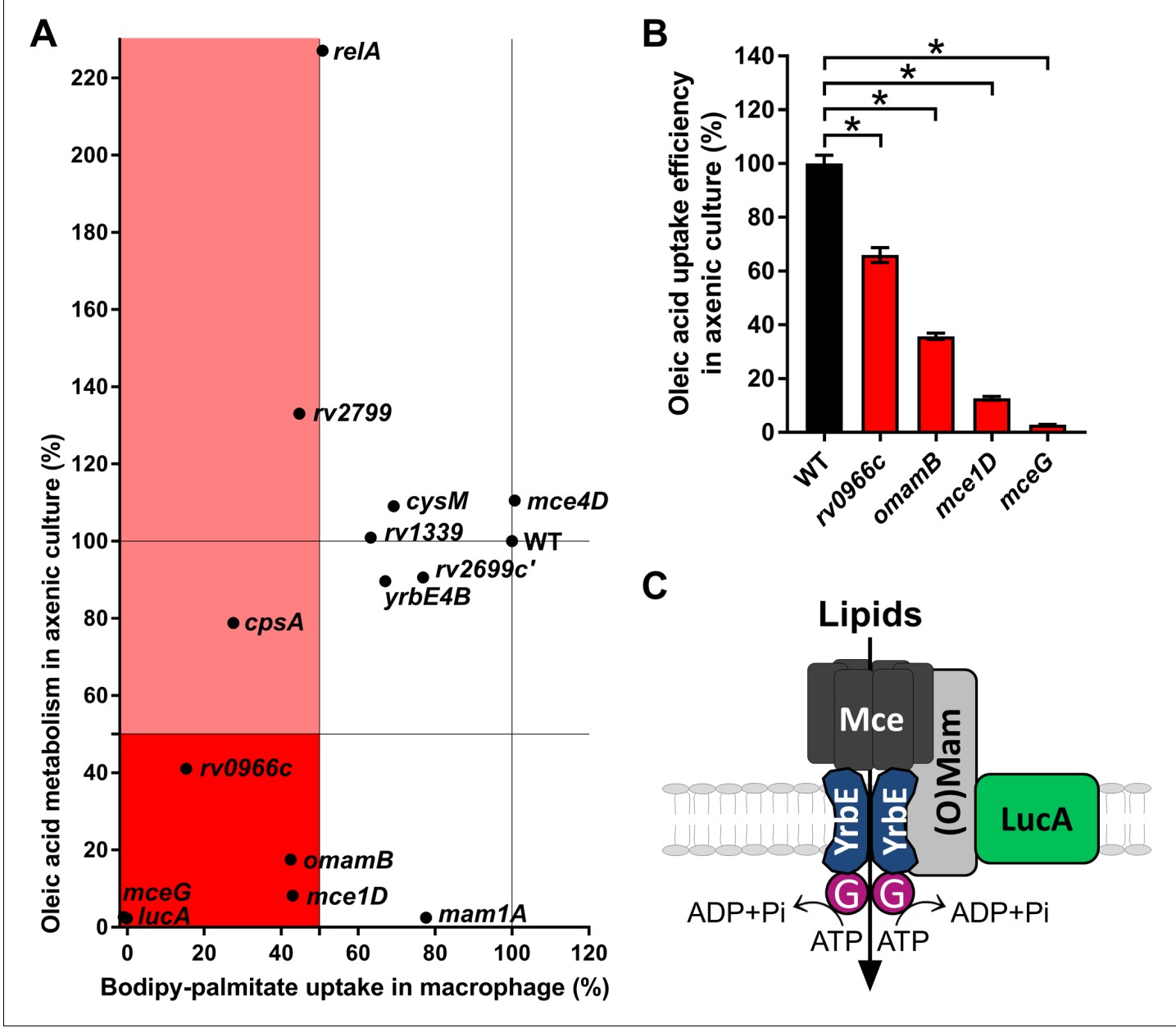

**Figure 3.** Phenotypic analysis of selected mutants. (**A**) Bodipy-palmitate incorporation by Mtb during macrophage infection in the selected mutants (X axis) and catabolic metabolism of $[1-^{14}C]$-oleic acid in axenic culture (Y axis). (**B**) Uptake of $[1-^{14}C]$-oleic acid by selected mutants in axenic culture. Data are means ± SD (n = 4). *p < 0.0001 (Student's t test). (**C**) Hypothetical model for Mce transporter configuration. Mce subunits are substrate specific and deliver lipids to the YrbE permease subunits. ATP is hydrolyzed by Rv0655/MceG (G) to transport substrates into the cytosol. LucA interacts with Mce transporters through accessory subunits Mam's or Omam's.

DOI: https://doi.org/10.7554/eLife.43621.007

The following figure supplement is available for figure 3:

**Figure supplement 1.** Confocal microscopy of selected mutants.
DOI: https://doi.org/10.7554/eLife.43621.008

the rate of fatty acid import. Transposon insertions in *rv0200/OmamB*, *rv0172/mce1D*, and *rv0655/mceG* had the most pronounced defect in fatty acid import (*Figure 3B*).

Together these data indicate that *rv0200/OmamB*, *rv0172/mce1D*, *rv0655/mceG*, and *rv0966c* are involved in fatty acid import in liquid culture and during macrophage infection. Alternatively,

transposon insertions in *rv0175/mam1A*, *rv2799*, *rv2583c/relA,* and *rv3484/cpsA* display condition-dependent reductions in the import and/or metabolism of fatty acids (*Figure 3A*).

## Discussion

The import of hydrophobic lipid substrates across the Mtb cell envelope is facilitated by a network of dedicated and shared proteins. Specifically, we proposed that Mce1 is a dedicated fatty acid transporter and LucA is a shared protein which interacts with Mce1 and Mce4 subunits to facilitate Mce1- and Mce4-mediated import of fatty acids and cholesterol, respectively (*Nazarova et al., 2017*). The link between LucA and the Mce1/Mce4 transporters suggests that fatty acid and cholesterol import is coordinated in Mtb and this may reflect the abundance of fatty acids and cholesterol within the macrophage niche. Thus, to build on our proposed model, we sought to identify additional proteins in this network that facilitate fatty acid import by Mtb during infection in macrophages.

Our model proposes that genes in the *mce1* operon encode the core subunits of a Mce1 fatty acid transporter. Consistent with this idea our screen selected numerous mutants with transposon insertions in the *mce1* operon. We identified and tested a representative mutant with an insertion in this operon (*rv0172/mce1D*) and confirmed that this mutant is defective in fatty acid import and metabolism of fatty acids (*Figure 3*). Our TraSH-based analyses also identified six total genes in the *mce1* operon that were overrepresented following the enrichment screen (*rv0168/yrbE1B*, *rv0169/mce1A*, *rv0171/mce1C*, *rv0172/mce1D*, *rv0175/mam1A*, and *rv0176/mam1B* (*Figure 2—source data 1*). This analysis also revealed that mutations in genes encoding the orphaned Mce accessory subunits (*rv0199/omamA* and *rv0200/omamB*) were overrepresented in our enrichment screen. Consistent with this prediction, a transposon mutant with an insertion in *rv0200/omamB* has a reduced ability to import and metabolize fatty acids (*Figure 3*). The accessory subunit, Rv0199/OmamA is known to interact with Rv3723/LucA (*Nazarova et al., 2017*) and to stabilize the Mce1 complex (*Perkowski et al., 2016*). It is possible that OmamB and OmamA function jointly to stabilize Mce1 through interactions with Rv3723/LucA. It has long been hypothesized that the putative ATPase, Rv0655/MceG provides energy for all Mce transporters in Mtb (*Joshi et al., 2006*). Rv0655/MceG has been shown to be required for cholesterol metabolism in Mtb (*Pandey and Sassetti, 2008*), and however, in *Mycobacterium smegmatis* (*García-Fernández et al., 2017*), its role in fatty acid assimilation had not been described previously. Among the mutants tested here *rv0655/mceG* had the strongest defect in fatty acid import and metabolism (comparable to *rv3723/lucA*), supporting its essential role in these processes (*Figure 3*). Thus, given that we identified a number of Mce1 subunits in the screen, we conclude that Mce1 functions as a fatty acid transporter within the macrophage environment (*Figure 3C*).

This screen also identified genes that had not been previously linked to fatty acid import in Mtb. Mutant with an insertion in *rv0966c* has a reduced ability to import Bodipy-palmitate during macrophage infection and this mutant also has a reduced ability to import and metabolize fatty acids in liquid culture (*Figure 3*). The putative protein encoded by *rv0966* contains a domain of unknown function (DUF1707) that is commonly found in proteins from Actinomycetal species. The gene *rv0966c* is highly expressed in a mouse infection model (*Talaat et al., 2007*) but more work will be needed to determine how Rv0966c facilitates fatty acids import in Mtb.

Inactivation of Rv2583c/RelA dramatically increases fatty acid metabolism in Mtb, yet this mutant imports less Bodipy-palmitate during macrophage infection (*Figure 3A*). Rv2583c/RelA mediates the stringent response following nutrient deprivation by regulating levels of (p)ppGpp (*Avarbock et al., 1999*). RelA is required for long-term survival of Mtb in vitro and in vivo; however, deletion of RelA is reported to have no effect on Mtb growth during short-term macrophage infections (*Dahl et al., 2003*; *Klinkenberg et al., 2010*; *Primm et al., 2000*; *Weiss and Stallings, 2013*). Inactivation of RelA decreases the expression of *mce1*, *mce3* and *mce4* operons in response to starvation (*Dahl et al., 2003*), suggesting that Rv2583c/RelA might be involved in the regulating lipid import in an environment-dependent manner.

We recovered mutants with transposon insertions in *rv3484/cpsA* from both rounds of enrichment and this mutation reduces Bodipy-palmitate import during macrophage infection (*Figure 3A*). The CpsA (capsular polysaccharide biosynthesis) protein participates in the ligation of arabinogalactan to peptidoglycan in the Mtb cell wall (*Grzegorzewicz et al., 2016*; *Harrison et al., 2016*). Within

macrophages, bacteria lacking CpsA also fail to block LC3-associated lysosomal trafficking, instead these bacteria traffic into NADPH-oxidase-positive phagosomes for clearance by the host cell (*Köster et al., 2017*). Thus, it is possible that the CpsA mutant's defect in Bodipy-palmitate import in macrophages may be due perturbations in the bacterial cell wall, intracellular trafficking and/or a decrease in bacterial viability.

This study describes a genetic screen to identify fatty acid import mutants in Mtb. Although we focused on fatty acid import here, it is understood that a balanced utilization of cholesterol and fatty acids is significant for Mtb fitness, growth and virulence (*Lee et al., 2013*; *Nazarova et al., 2017*; *Wilburn et al., 2018*). Our ongoing work is now focused on characterizing the proteins that coordinate fatty acid and cholesterol import by Mtb. We hypothesize that fatty acid and cholesterol import in Mtb are governed by a network of proteins that facilitate metabolic adaptations to various environmental pressures that the bacteria encounter during infection (*Huang et al., 2018*). Growing evidence indicates that Mtb's metabolism is constrained within macrophages (*VanderVen et al., 2015*) yet is integrated to the metabolism of host cells. Thus, understanding the complex process of lipid import in Mtb during infection may shed new light on weakness in the bacterial adaptation to host environment that can be targeted with antituberculosis therapies.

## Materials and methods

All Materials and methods are described (*Nazarova et al., 2017*), unless specified otherwise.

### Flow cytometry and sorting

Bacteria were pre-grown to mid log phase in 7H9 OAD media. Murine bone marrow-derived macrophages were seeded into T-75 tissue culture flasks ($2 \times 10^7$ cells per flask) and infected with Mtb at a MOI of 4:1. Four flasks were infected with mutant library for each round of the screen. In parallel, on each day of experiment macrophages were infected with WT and ΔlucA::hyg to establish sorting gates. At day three of infection Bodipy-palmitate (final concentration 9 µM) conjugated to de-fatted 1% BSA (in PBS) was added to the cells for one hour pulse. Labeled cells were chased with fresh cell media for 1 more hour. Isolation of intracellular bacteria was performed as described previously (*Nazarova et al., 2018*). Isolated bacteria were washed twice in PBS + 0.05% tyloxapol, and once in PBS + 0.1% Triton X-100 +0.1% fatty acid-free BSA. Bacteria were suspended in PBS + 0.05% tyloxapol and passed through a 25-gauge needle 20 times. The cell suspensions were then diluted in Basal uptake buffer (*Nazarova et al., 2018*) for flow cytometry analysis or sorting, which were performed on S3 sorter (Bio-Rad). The percentage of Bodipy-palmitate import by the individual mutants (Tn) was determined by normalization of mean fluorescence (MF) using following formula, where 100% of uptake was assigned to WT, and 0% - to Δ*lucA*::hyg mutant (LucA).

Bodipy-palmitate uptake (%) = $(MF_{Tn} - MF_{LucA})/(MF_{WT} - MF_{LucA}) \times 100\%$

### Transposon mutant screen

A MycoMarT7 transposon (kan$^R$) mutant mutant library (~$2 \times 10^4$ independent mutants) was generated as described (*Lee et al., 2013*) in a wild type Mtb Erdman background that has the chromosomally integrated plasmid, pDEA43n *smyc'::mCherry* (hyg$^R$). Sorting was performed as described above, and sorted events were plated onto 7H10 OAD agar (for both sorting rounds) containing kanamycin (25 ug/ml) and hygromycin (100 ug/ml). Pooled mutants after the first round of screen were frozen in 15% glycerol and re-grown in 7H9 OAD to mid log phase before re-infection of macrophages for the second enrichment round. DNA from individual mutants were picked from plates and recovered in 7H9 OAD culture for both rounds. Individual mutants was isolated and the transposon insertion sites were PCR amplified and sequenced as described (*Prod'hom et al., 1998*).

### Transposon site Hybridizationhybridization (TraSH)

The MycoMarT7 transposon carries two outward-facing T7 promoter elements and to map transposon insertions across the entire genome dye-labeled RNA is generated and hybridized to a custom M. tuberculosis specific tiling microarray. To do this, genomic DNA from the mutant pools was extracted as described (*Lee et al., 2013*; *Murry et al., 2008*) with minor modifications. Genomic DNA from each pool was partially digested separately with BstNI ($2 \times 100$ µl reaction mixtures each containing 10 µl 10 mg/ml BSA, 30 µg DNA, 10 µl NEBuffer 2 and 40 units of enzyme, and with SfaNI

(6 × 100 µl reaction mixtures each containing 10 µl 10 µg DNA, 10 µl NEBuffer 3 and 20 units of enzyme (New England Biolabs). The target 0.5–2 kb fragmentsof genomic DNA were size selected using a 0.7% agarose gel and purified with QIAquick gel extraction kit (Qiagen) followed by an additional isopropanol precipitation to remove agarose contaminants. 500 ng of each digested genomic DNA sample werecombined into one*in vitro* transcription with the MEGAshortscript T7 Transcription Kit (Ambion) with 1 µg DNA total per reaction. The *in vitro* transcription reactions incorporated amino-allyl UTP as descrribed (*Rohde et al., 2007*) Details of RNA labeling, purification, and array hybridization were conducted as described (*Liu et al., 2016*). Genespring GX, Agilent software was used to analyze the TraSH data. Raw data for TraSH analysis is deposited in the NCBI Gene Expression Omnibus database accession number (GSE119753) (*Edgar et al., 2002*).

## Lipid uptake assay

Lipid uptake was quantified as described previously with slight modifications. Briefly, Mtb was cultured in 7H9 AD medium in vented standing T-75 tissue culture flasks. After 5 days, cultures were normalized to $OD_{600}$ of 0.7 in 8 ml using spent medium, and 0.2 µCi of [1-$^{14}$C]-oleate (Perkin Elmer) was added to bacteria. After 5, 30, 60, and 120 min of incubation at 37°C 1.5 ml of the bacterial cultures were collected by centrifugation, washed three times in 1 ml of ice-cold wash buffer (0.1% Fatty acid free-BSA and 0.1% Triton X-100 in PBS), fixed in 0.2 ml of 4% PFA for 1 hr. Bacterial associated radioactivity was quantified by scintillation counting. with slight modifications.

## Lipid metabolism assay

Lipid metabolism was monitored by quantifying the release of $^{14}CO_2$ from [1-$^{14}$C]-oleate by radiorespirometry as described previously. Briefly, Mtb were grown in 7H9 AD for 5 days, the culture was adjusted to an $OD_{600}$ of 0.7 in 5 ml spent medium containing with 1.0 µCi of [1-$^{14}$C]-oleate in vented standing T-25 tissue culture flask. These flasks were then placed in a sealed air-tight vessel with an open vial containing 0.5 ml 1.0 M NaOH at 37°C. After 5 hr, the NaOH vial was recovered, neutralized with 0.5 ml 1.0 M HCl, and the amount of base soluble $Na_2^{14}CO_3$ was quantified by scintillation counting.

## Acknowledgements

We thank Linda Bennett for excellent technical support. This work was supported by the National Institute of Health (AI118582 and AI134183) to DGR and (AI099569 and AI119122) to BCV.

## Additional information

### Funding

| Funder | Grant reference number | Author |
| --- | --- | --- |
| National Institutes of Health | AI099569 | Brian C VanderVen |
| National Institutes of Health | AI119122 | Brian C VanderVen |
| National Institutes of Health | AI118582 | David Russell |
| National Institutes of Health | AI134183 | David Russell |

The funders had no role in study design, data collection and interpretation, or the decision to submit the work for publication.

### Author contributions

Evgeniya V Nazarova, Conceptualization, Data curation, Formal analysis, Validation, Investigation, Visualization, Methodology, Writing—original draft, Writing—review and editing; Christine R Montague, Lu Huang, Thuy La, Investigation, Methodology; David Russell, Conceptualization, Resources, Supervision, Funding acquisition, Project administration, Writing—review and editing; Brian C VanderVen, Conceptualization, Resources, Supervision, Funding acquisition, Investigation, Methodology, Project administration, Writing—review and editing

Author ORCIDs
Lu Huang http://orcid.org/0000-0002-7820-2177
David Russell http://orcid.org/0000-0002-9748-750X
Brian C VanderVen http://orcid.org/0000-0003-3655-4390

Ethics
Animal experimentation: This study was performed in strict accordance with the recommendations in the Guide for the Care and Use of Laboratory Animals of the National Institutes of Health. All of the animals were handled according to approved institutional animal care and use committee (IACUC) protocols (2013-0030) from Cornell University.

Decision letter and Author response
Decision letter https://doi.org/10.7554/eLife.43621.014
Author response https://doi.org/10.7554/eLife.43621.015

## Additional files

### Supplementary files
• Transparent reporting form
DOI: https://doi.org/10.7554/eLife.43621.009

### Data availability
Data have been deposited in GEO under accession code GPL25556.

The following dataset was generated:

| Author(s) | Year | Dataset title | Dataset URL | Database and Identifier |
|---|---|---|---|---|
| Nazarova E | 2018 | Array for TRASH screen analysis | https://www.ncbi.nlm.nih.gov/geo/query/acc.cgi?acc=GPL25556 | NCBI Gene Expression Omnibus, GPL25556 |

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
