## [Decision Letter]

Thank you for submitting your article "The genetic requirements of fatty acid import by *Mycobacterium tuberculosis* within macrophages" for consideration by *eLife*. Your article has been reviewed by three peer reviewers, one of whom is a member of our Board of Reviewing Editors, and the evaluation has been overseen by Gisela Storz as the Senior Editor. The following individuals involved in review of your submission have agreed to reveal their identity: William R Jacobs (Reviewer #2); Helena Boshoff (Reviewer #3).

The reviewers have discussed the reviews with one another and the Reviewing Editor has drafted this decision to help you prepare a revised submission.

Summary:

It is now well accepted that *Mycobacterium tuberculosis*, the causative agent of tuberculosis, is able to utilize host-derived carbon sources, including those obtained from fatty acids. The manuscript by Nazarova, Vanderven and colleagues explores this further and builds on previous work published by these authors that describe the role of LucA in stabilizing fatty acid transport systems. In this submission, the authors develop a forward genetic screen to identify genes required for fatty acid uptake. The system involves the use of a fluorescently labelled fatty acid derivative, combined with a transposon library and flow cytometry. They validate this system and appending gating strategy using the previously reported lucA mutant, which accumulates lower levels of fluorescent fatty acid when compared to the wild type.

Key findings:

1) Screening of a transposon library of *M. tuberculosis* in this optimized assay revealed a couple of known associations with lipid import but also revealed a few novel genes that are defective in fatty acid import.

2) The role of some genes that were predicted to be part of the Mce1 transporter or part of the *mce1* operon are confirmed whereas Rv0996c, a gene of unknown function, is found to play a role in lipid uptake.

3) The lipid metabolism role of several of the genes identified in the screen is further confirmed by measurement of β-oxidation of exogenously added oleic acid.

Concerns:

1) The Abstract is too diffuse and whilst it conveys the methodological approach and broad swathes of data, some specifics are needed. Perhaps mention some specific genes.

2) Figure 3—figure supplement 1 needs some form of quantification and graphic representation. It is difficult to see the differences pictorially.

3) The authors should review their citations as the key paper, of their own, is cited incorrectly.

4) Although they take advantage of wonderful tools such as transposon mutagenesis and integration proficient vectors, they do not cite the original studies in which these were described and on which their work is dependent on.

---

## [Author Response]

Concerns:1) The Abstract is too diffuse and whilst it conveys the methodological approach and broad swathes of data, some specifics are needed. Perhaps mention some specific genes.2) Figure 3—figure supplement 1 needs some form of quantification and graphic representation. It is difficult to see the differences pictorially.3) The authors should review their citations as the key paper, of their own, is cited incorrectly.4) Although they take advantage of wonderful tools such as transposon mutagenesis and integration proficient vectors, they do not cite the original studies in which these were described and on which their work is dependent on.

Based on the previous reviewer’s comments we have made the following changes. We streamlined the Abstract to a more concise form, the visual data in Figure 3—figure supplement 1 has now been quantified and depicted in a new panel including statistics, and we have properly cited the key work which developed the genetic tools that we used to manipulate *M. tuberculosis* that we used here.